# Early Life Interventions for Childhood Growth and Development in Tanzania (ELICIT): a protocol for a randomised factorial, double-blind, placebo-controlled trial of azithromycin, nitazoxanide and nicotinamide

Mark Daniel DeBoer,[1] James A Platts-Mills,[2] Rebecca J Scharf,[1,2] Joann M McDermid,[2] Anne W Wanjuhi,[1] Jean Gratz,[2] Erling Svensen,[3] Jon R Swann,[4] Jeffrey R Donowitz,[5] Samwel Jatosh,[6] Eric R Houpt,[2] Estomih Mduma[6]

For numbered affiliations see end of article.

**Correspondence to**
Dr Mark Daniel DeBoer;
deboer@virginia.edu

## ABSTRACT

**Introduction** In many developing areas in the world, a high burden of enteric pathogens in early childhood are associated with growth deficits. The tryptophan-kynurenine-niacin pathway has been linked to enteric inflammatory responses to intestinal infections. However, it is not known in these settings whether scheduled antimicrobial intervention to reduce subclinical enteric pathogen carriage or repletion of the tryptophan-kynurenine-niacin pathway improves linear growth and development.

**Methods and analysis** We are conducting a randomised, placebo-controlled, factorial intervention trial in the rural setting of Haydom, Tanzania. We are recruiting 1188 children within the first 14 days of life, who will be randomised in a 2×2 factorial design to administration of antimicrobials (azithromycin and nitazoxanide, randomised together) and nicotinamide. The nicotinamide is administered as a daily oral dose, which for breast-feeding children aged 0–6 months is given to the mother and for children aged 6–18 months is given to the child directly. Azithromycin is given to the child as a single oral dose at months 6, 9, 12 and 15; nitazoxanide is given as a 3-day course at months 12 and 15. Mother/child pairs are followed via monthly in-home visits. The primary outcome is the child's length-for-age Z-score at 18 months. Secondary outcomes for the child include additional anthropometry measures; stool pathogen burden and bacterial microbiome; systemic and enteric inflammation; blood metabolomics, growth factors, inflammation and nutrition; hydrogen breath assessment to estimate small-intestinal bacterial overgrowth and assessment of cognitive development. Secondary outcomes for the mother include breastmilk content of nicotinamide, other vitamins and amino acids; blood measures of tryptophan-kynurenine-niacin pathway and stool pathogens.

**Ethics and dissemination** This trial has been approved by the Tanzanian National Institute for Medical Research, the Tanzanian FDA and the University of Virginia IRB.

## Strengths and limitations of this study

► This study uses a factorial design to determine independent contributions of two different intervention approaches on early childhood growth and development.
► The study is set in a rural site in sub-Saharan Africa with a high degree of both enteric pathogen carriage and stunting.
► This study assesses intervention with nicotinamide, targeting the tryptophan-kynurenine-niacin pathway, which may have important roles in intestinal bacterial content and immune and metabolic response.
► In addition to the primary outcome of length-for-age Z-score, this study measures extensive secondary outcomes related to infection, nutrition, metabolism and cognitive development.
► Due to sample size constraints, one limitation of the study is that two different antimicrobial interventions (azithromycin and nitazoxanide) are randomised together, limiting the ability to determine any individual contribution of either of these to study outcomes.

Findings will be presented at national and international conferences and published in peer-review journals.
**Protocol version** 5.0, 4 December 2017.
**Protocol sponsor** Haydom Lutheran Hospital, Haydom, Manyara, Tanzania.
**Trial registration number** NCT03268902; Pre-results.

## INTRODUCTION

Childhood growth and thriving remain suboptimal in low-resource settings worldwide, underscoring the need to better

understand underlying aetiologies and to identify interventions towards improving outcomes.[1 2] The Etiology, Risk Factors, and Interactions of Enteric Infections and Malnutrition and the Consequences for Child Health (MAL-ED) study evaluated correlates of childhood growth and cognitive development in eight low-income to middle-income countries worldwide, including the region around Haydom, Tanzania.[3 4] Haydom is in a rural, semiarid area with a high degree of poverty. Children in Haydom followed in MAL-ED had a prevalence of stunting of 70% at 18 months of age.[5] Data from MAL-ED also showed that linear growth deficits correlated with lower scores for cognitive development,[6] potentially through shared barriers to optimal growth and brain development.[1 7]

### Antimicrobials to improve childhood growth

In MAL-ED, children in the Haydom site had a high burden of infection with intestinal pathogens starting early in life. High carriage of enteropathogens, even in the absence of diarrhoea, was associated with a 1.36 increased odds of having lower length for age.[3] As for the specific enteropathogens, among the most prevalent pathogens at age 6 months were enteroaggregative *E. coli* (48%) and *Campylobacter* (27%),[8] each of which were associated with a 0.85 cm and 0.83 cm, respectively, shortfall in height attainment.[9] Both of these infections are treatable with azithromycin. Other highly prevalent pathogens by age 6 months in the MAL-ED study included *Giardia* (22%) and

*Cryptosporidium* (6%), and early *Giardia* infection was also associated with growth shortfalls.[10] Both of these protozoal infections are treatable with nitazoxanide. These observations raise questions as to whether such antibiotics may be useful in ameliorating growth deficits.

A meta-analysis of 10 randomised controlled trials showed that antibiotic use increased height by 0.04 cm/month.[11] Another study demonstrated that randomised administration of azithromycin to asymptomatic young children in Ethiopia was associated with a halving of mortality,[12] suggesting overall effects of suppressing bacterial pathogens in areas with high rates of infectious disease. However, results from individual studies of azithromycin mass drug-administration on growth have not shown consistent results.[13] Among children with cryptosporidial infection in Zambia, treatment with nitazoxanide resulted in reduced mortality, while linear growth was not assessed.[14] Therefore, the role of these antimicrobials for linear growth remains unclear.

### Nicotinamide to improve childhood growth

Another potential means of improving growth is by targeting the tryptophan-kynurenine-niacin pathway (figure 1). In this pathway, dietary tryptophan is absorbed and then metabolised by indoleamine 2,3-dioxygenase (IDO) to kynurenine and later to nicotinic acid, a key precursor to nicotinamide adenine dinucleotide (NAD+). The importance of this pathway has been supported by recent work identifying mutations in the NAD+ pathway as

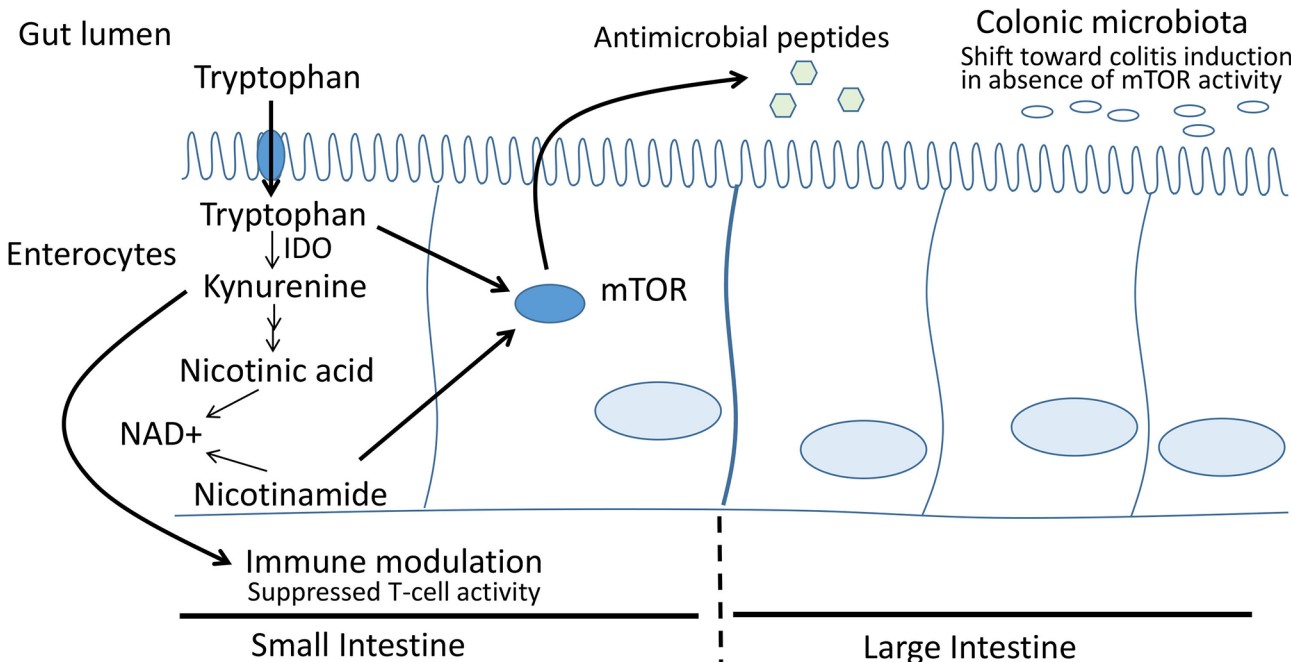

**Figure 1** The tryptophan-kynurenine-niacin pathway in the intestinal mucosa. Dietary tryptophan is metabolised to kynurenine by IDO—whose activity is increased in settings of inflammation. Adverse effects of elevated IDO activity relate to modulation of the immune response, decreased activity of mTOR in producing antimicrobial peptides and decreased tryptophan for protein synthesis. Intervention with nicotinamide ameliorates some of these effects in preclinical models; in the current study, we hypothesise that intervention with nicotinamide will increase linear growth via normalisation of the tryptophan-kynurenine-niacin pathway and restored enterocyte activity. Both nicotinic acid and nicotinamide are considered dietary forms of niacin. IDO, indoleamine 2,3-dioxygenase; NAD, nicotinamide adenine dinucleotide.

a cause of birth defects in humans; in animal models with these mutations, birth defects were prevented via supplementation with niacin.[15] Deficiencies in this pathway have been noted in areas where diets are heavily maize-based, leading to pellagra. Infants at the Haydom site in MAL-ED were noted to have inadequate dietary intake of multiple vitamins[16]; while deficiency in niacin was not assessed, individuals in the region around Haydom eat a diet based heavily on maize, with potentially low bioavailability of tryptophan. Basic science models have demonstrated that tryptophan deficiency is associated with reduced growth velocity.[17]

Abnormalities in the tryptophan-kynurenine-niacin pathway are also associated with intestinal inflammation, including during response to intestinal pathogens.[18 19] In animal models, an inability to transport tryptophan into enterocytes results in diarrhoea, while provision of nicotinamide (a form of niacin) partly restores function of mTOR in enterocytes, increasing secretion of antimicrobial peptides and reducing diarrhoea,[18 19] suggesting effects on diarrheal pathogens. IDO activity is increased during inflammation, including in inflammatory bowel disease[20 21] and during infections[22]; this appears to be a maladaptive in most cases, as kynurenine has potential immunosuppressive effects.[23 24] In MAL-ED, children at the Haydom site with higher ratios of kynurenine:tryptophan were noted to have poorer linear growth.[24] Given that (1) children in the Haydom area have a maize-based diet low in tryptophan, a key precursor to niacin/nicotinamide and NAD+, (2) preclinical studies demonstrate improvements in antimicrobial peptides and reduced diarrhoea with nicotinamide supplementation, (3) an increase in nicotinamide as a precursor of NAD+ should cause reduced need for the tryptophan-kynurenine-niacin pathway, reducing activity of IDO and thereby reducing kynurenine and (4) higher kynurenine:tryptophan is associated with poor growth, we reasoned that intervention with nicotinamide among children in this region would improve linear growth. However, because growth is a long-term process and nicotinamide is a water-soluble vitamin without a mechanism for ongoing storage, we reasoned that an intervention with nicotinamide would need to be given over a long period of time.

### Study hypotheses

We hypothesised that among children with a high degree of stunting and enteric pathogen burden in this region, intervention with (1) antimicrobials and/or (2) nicotinamide would increase linear growth. We further set out to interrogate the potential mechanisms behind these effects. In the case of antimicrobial intervention, we hypothesise that intervention with azithromycin and nitazoxanide (relative to placebo) would result in a decrease in enteropathogen burden that would be associated with an increase in growth. In the case of nicotinamide (relative to placebo), we hypothesise that intervention will result in a reduction in the kynurenine:tryptophan ratio, an increase in availability of nicotinamide for NAD+ synthesis and associated increases in growth.

### OBJECTIVE

The primary objectives of this randomised factorial, double-blind, placebo-controlled trial are to determine if interventions with (1) antimicrobials and (2) nicotinamide are associated with increased length-for-age Z-score (LAZ, relative to placebo-treated children) by age 18 months. Specifically, the antimicrobial intervention consists of azithromycin administered as a single dose at age 6, 9, 12 and 15 months and nitazoxanide as a 3-day course at age 12 and 15 months (with both antimicrobials randomised together). The nicotinamide intervention consists of nicotinamide 250 mg daily administered daily by mouth to the child's mother during breast feeding 0–6 months (and thus passed on to the child via breast milk[25]) and then 100 mg administered daily by mouth to the child during months 6–18. These interventions are delivered in a factorial manner such that children receive both interventions, either intervention alone or neither intervention.

Because multiple prior large-scale studies of other preparations of vitamins and antibiotics have failed to show effects on linear growth, we have multiple secondary outcomes to interrogate additional physiological effects of the interventions—potentially helping to identify underlying mechanisms. These secondary outcomes include measures of inflammation, intestinal bacterial content (pathogens and otherwise) and measures of metabolism, development and health.

### SETTING

Haydom is a rural area in north-central Tanzania, approximately 300 km west of Arusha, the nearest urban centre.[4] There is poor infrastructure for roads and other resources, and the local population of approximately 20 000 depends primarily on subsistence agriculture. Haydom Lutheran Hospital (HLH) is the central hub for the area and has a strong relationship with the Haydom community. Healthcare in the area relies both on HLH and reproductive child health system (RCHS) with 38 outreach clinics across 74 villages, where children are followed monthly for weight-checks and immunisations. Given the low HIV seroprevalence (<2%) among mothers of childbearing age, HIV testing is not performed in the current study. Because of its elevation (1700 m), Haydom is not a high-risk area for malaria, with almost no *Plasmodium falciparum* in the region. Among the families followed in MAL-ED, 92.3% lived in houses with mud floors, 0% had electricity and 67.9% obtained their water from an unprotected well or surface water.[4] Further estimates of the low socioeconomic status of the area have been published previously.[4]

## METHODS AND ANALYSIS (STANDARD PROTOCOL ITEMS FOR RANDOMISED TRIALS)

### Eligibility

Participants are eligible if the mother is ≥18 years and the child is age ≤14 days and was born in the Haydom catchment area (defined for study purposes as 25 km radius of Haydom Lutheran Hospital).

Exclusion criteria are as follows: maternal inability to adhere to the protocol, multiple gestation, significant birth defect or neonatal illness, weight <1500 g at enrolment (age ≤14 days), lack of intent to breastfeed infant and plan to move from area within the next 18 months.

### Recruitment

Recruitment occurs via survey by field teams of surrounding villages and via input from local community healthcare workers. Pregnant women are identified and informed regarding the study. Pregnant women in the third trimester may provide informed consent for the study but are not enrolled until the initial assessment visit of the infant to determine eligibility (which includes ascertainment of infant weight). If informed consent is not obtained during pregnancy, it is obtained at enrolment. The general timeline of study procedures is shown in table 1.

### Enrolment

Infant/mother pairs are enrolled during the initial study visit to their homes between birth and day of life 14, after eligibility criteria are confirmed, including measurement of infant weight which should be 1.5 or above. Informed consent is provided at this time, if it was not obtained during pregnancy. Eligibility (besides current weight) is confirmed and the mother/child pair is randomised in the study.

### Randomisation

Randomisation occurs at the time of enrolment. Block randomisation was used in permuted blocks of eight for the four intervention combinations (antimicrobial+nicotinamide, antimicrobial alone, nicotinamide alone, all placebo). Study teams are assigned permuted blocks before initial study visits to sequentially allocate randomisation status to participants based on study identification number assigned prior to the initial study visit.

### Intervention

As a factorial study design, the two intervention domains (antimicrobial and nicotinamide) are randomised separately. The antimicrobial intervention consists of two medications that are randomised together: azithromycin and nitazoxanide (versus placebos of both). Azithromycin 20 mg/kg (vs placebo) (both manufactured by Universal Corp, Kenya) is administered by study personnel at months 6, 9, 12 and 15 as a single dose of azithromycin (200 mg/5 mL suspension by mouth). This timing was selected (versus earlier initiation) because while children have enteropathogen carriage before 6 month in this area,[8] carriage of azithromycin-susceptible pathogens

such as *Campylobacter* are not highly prevalent, and the decline in LAZ is steeper in the 6–18-month window compared with the 0–6-month window.[5] The nitazoxanide (versus placebo) (both manufactured by Romark, Florida, USA) is administered as a course of nitazoxanide (100 mg/5 mL) 100 mg two times per day for 3 days. This timeline was selected because nitazoxanide is not approved for use below 12 months. The first of these nitazoxanide doses is administered by the study personnel and the remaining doses are administered at home by the child's family. Families are asked at a subsequent visit about whether they completed the course of nitazoxanide.

The nicotinamide (versus placebo) (both manufactured by Vita-gen, New York, USA) is delivered as a daily dose throughout the study. During the first 6 months of life, this is delivered to the child via breast-milk. This is performed because of the lack of availability of a liquid preparation of nicotinamide and to emphasise the importance of exclusive breast feeding (ie, that intake beyond breast milk is not needed during the first 6 months of life). Mothers take nicotinamide 250 mg tablets, one tablet by mouth daily. Prior studies have demonstrated that nicotinamide is present in breast milk in quantities that are associated with serum levels.[25] Haydom-specific data from MAL-ED demonstrated a high prevalence (96%) of breast feeding until children were 6 months old.[5] Because the prevalence of exclusive breastfeeding was low in the area around Haydom during MAL-ED, the current study includes a breast-feeding encouragement and support initiative, using nurses who have received specialised training in breast feeding; these nurses work with the local community healthcare workers in home visits to participating families in the first 6 months of the study. Their message is that exclusive breast feeding is better for mother and baby (highlighting a variety of reasons) and that if the family chooses not to breast feed, that providing breast milk before any complimentary foods will allow the child to receive a greater amount of the needed nutrition—including a greater amount of the nicotinamide ingested by the mother. The lack of intent to breast feed is an exclusion criterion at the time of enrolment; however, if the mother dies or discontinues breast feeding, the child subsequently would not receive nicotinamide/placebo until age 6 months.

From months 6–18, the child receives sachets of nicotinamide 100 mg by mouth daily. Mothers are instructed to mix the nicotinamide powder with a small volume of age-appropriate food (eg, infant cereal) to give to the child.

For both the maternal nicotinamide pills and the children's nicotinamide sachets, doses are given as a 60 day supply every 2 months. When a new supply is provided, the family is asked for the remainder of doses not consumed the prior 60 days. These unused doses are counted and recorded by pharmacy staff to determine adherence.

**Table 1** Timing of interventions and measurements

| | Pregnancy | Birth-2 weeks | 1 | 2 | 3 | 4 | 5 | 6 | 7 | 8 | 9 | 10 | 11 | 12 | 15 | 18 |
|---|---|---|---|---|---|---|---|---|---|---|---|---|---|---|---|---|
| | | | \multicolumn Month of life | | | | | | | | | | | | | |
| Recruitment, consent | X | X | | | | | | | | | | | | | | |
| Enrolment | | X | | | | | | | | | | | | | | |
| **Interventions** | | | | | | | | | | | | | | | | |
| A. Nicotinamide intervention | | | | | | | | | | | | | | | | |
| Nicotinamide (or placebo) to mother: daily 0–6 months | | | X | X | X | X | X | X | | | | | | | | |
| Nicotinamide (or placebo) to child: daily 6–12 months | | | | | | | X | X | X | X | X | X | X | X | | |
| B. Antimicrobial intervention | | | | | | | | | | | | | | | | |
| Azithromycin single dose to child | | | | | | | | X | | | X | | | X | X | |
| Nitazoxanide 3 day course to child | | | | | | | | | | | | | | X | X | |
| **Measurements** | | | | | | | | | | | | | | | | |
| A. Mother | | | | | | | | | | | | | | | | |
| Weight: Month 11 | | | | | | | | | | | | | X | | | |
| Height: Month 11 | | | | | | | | | | | | | X | | | |
| Blood | | | | | | | | X | | | | | | | | |
| Stool | | | | | | | | X | | | | | | | | |
| Saliva (FUT2 secretor status) | | | | | | | | | X | | | | | X | | |
| B. Child | | | | | | | | | | | | | | | | |
| 1. Anthropometry | | | | | | | | | | | | | | | | |
| Weight | | X | X | X | X | X | X | X | X | X | X | X | X | X | X | X |
| Length, head circumference, mid-upper arm circ. | | X | | | X | | | X | | | X | | | X | X | X |
| 2. Questionnaire | | | | | | | | | | | | | | | | |
| Education, environment, socioeconomic | | | X | | | | | | | | | | | | | |
| Illness, treatment, breast feeding, food security | | | X | X | X | X | X | X | X | X | X | X | X | X | X | X |
| 3. Breast milk: Months 1 and 5 | | | | | | | | | | | | | | | | |
| Tryptophan, niacin, nicotinamide, vitamers | | | X | | | | X | | | | | | | | | |
| Vitamins | | | X | | | | X | | | | | | | | | |
| Protein | | | X | | | | X | | | | | | | | | |
| 4. Blood | | | | | | | | | | | | | | | | |
| Safety assessment (CMP, CBC in subset) | | | | X | | | | | | X | | | | | | |
| Haemoglobin | | | | | | | | | | | | | | X | | X |
| IGF-1 | | | | | | | | | | | | | | X | | X |
| CRP | | | | | | | | | | | | | | X | | X |
| Collagen X | | | | | | | | | | | | | | X | | X |
| Tryptophan metabolites | | | | | | | | | | | | | | X | | X |
| Metabolomics | | | | | | | | | | | | | | X | | X |
| 5. Stool | | | | | | | | | | | | | | | | |
| Microbiome | | | | | | | | X* | | | | | | X | | X |
| Taqman Array Cards | | | | | | | | X* | | | | | | X* | | X |
| Myeloperoxidase | | | | | | | | X | | | | | | X | | X |
| 6. Urine | | | | | | | | | | | | | | | | |
| Metabolomics | | | | | | | | X | | | | | | X | | X |
| 7. Saliva | | | | | | | | | | | | | | | | |

Continued

**Table 1** Continued

| | | Birth- | Month of life | | | | | | | | | | | | | | |
|---|---|---|---|---|---|---|---|---|---|---|---|---|---|---|---|---|---|
| | Pregnancy | 2 weeks | 1 | 2 | 3 | 4 | 5 | 6 | 7 | 8 | 9 | 10 | 11 | 12 | 15 | 18 |
| FUT2 secretor status | | | | | | | | X | | | | | | | | |
| 8. Breath | | | | | | | | | | | | | | | | |
| H$_2$, methane (small intestinal bacterial overgrowth) | | | | | | | | X | | | | | | X | | X |
| 9. Developmental testing | | | | | | | | | | | | | | | | |
| Observation of Maternal Child Interaction | | | | | | | | | | | | | | X | | |
| Malawi Developmental Assessment Tool | | | | | | | | | | | | | | | | X |
| Bayley Scales of Development | | | | | | | | | | | | | | | | X |

*Evaluation also performed two weeks after anti-microbial dose.
CBC, complete blood count; CMP,comprehensive metabolic panel; CRP, C reactive protein; FUT2, fucosyltransferase 2; IGF-1, insulin-like growth factor-1; ITT, intention-to-treat.

## Data collection

### Questionnaires

Study staff use standardised questionnaires designed for the study, in many cases adapted from questionnaires used in the MAL-ED study. These include questions asked at each monthly visit regarding breastfeeding, use of complementary foods, childhood illness and treatment (including whether an antibiotic was used and if so which one), types of food sources given and food insecurity. Additional questionnaires are asked at other specific time periods, including aspects of the child's home environment (completed at the first month's visit) and the vaccines that the child has received (completed at the 12-month visit).

### Anthropometry measures

All field team members are trained in the accurate measurement of anthropometry indices. The child's weight is measured at the initial visit and at each monthly visit through age 18 months. All anthropometry measures for a given month (including the final visit) are able to be performed up to 14 days before or after the target date (which is based on the number of months after the child's birthday). The child's length and head circumference are measured at the initial visit and every three visits thereafter through 18 month. The mother's weight and height are measured at month 10–11. Quarterly, weights are measured using a digital scale that undergoes quality assessments weekly. Infants are weighed on an infant tray with the infant nude or wearing a dry diaper or standing, when able. In between these quarterly weight measures, weights are measured monthly using hanging spring scales that are zeroed before each use and assessed for accuracy weekly. Lengths are measured quarterly by two field team members with the child lying flat on a measuring board with a fixed, perpendicular board against which the head is placed and a second adjustable perpendicular board is placed against the feet. Two measurements are obtained; if these are within 2 mm, the average of the two is used; if these are not within 2 mm, a third measurement is obtained and the average of the two closest measurements is used. Head circumference is measured quarterly by placing a non-distensible tape around the child's head just above the eyebrows, above the ears and around the biggest part of the back of the head. Mid-upper arm circumference is measured quarterly by using the same measuring tape at the mid-point of the upper arm segment. Maternal weight is measured using a standing scale. Maternal height is measured using a portable stadiometer. For quality assurance, 5% of infant weights, lengths and head circumferences are measured by a supervisor skilled in measurement to compare with the field team's measurement.

### Developmental assessments

To assess maternal-child interaction, an important predictor of early childhood development and health,[26 27] The Observation of Maternal Child Interaction (OMCI)[28] assessment is collected at 12 months of age. This observation includes a 5 min observation of mother and child interacting over a picture book, with ratings given for positive affect, verbal statements, sensitivity, scaffolding, language stimulation, focus and mutual enjoyment, among others.

At 18 months, developmental assessment is performed by trained assessors using two standardised scores, the Malawi Developmental Assessment Tool (MDAT)[29] and the Bayley Scales of Infant and Toddler Development, cognitive assessment (Bayley).[30] The MDAT is a validated score that tests multiple childhood development domains, including gross motor, fine motor, language and social development that has been validated in this age range in Malawi and intended for use in rural sub-Saharan settings. Briefly, this assessment was designed to be a culturally relevant method for examining child development across multiple domains using objects and tools typically available in rural or lower-resource settings. Both direct assessment and caregiver report of skills go into the score.

The Bayley cognitive assessment involves a series of tasks presented to the infant, which assess for play, reasoning and mental representation. This is a commonly used assessment for early childhood and has been used in many research sites around the world.[6 31]

## Specimen collection
### Breast milk (months 1 and 5)
Participating mothers express a maximal volume of breast milk from one breast (up to 40 mL), using hand expression under low-light conditions. Breast milk will be collected into foil-covered sterile containers and transported on ice to the central laboratory centre within 8 hours for archiving of whole milk at −80°.

### Blood (months 12 and 18; in a subset months 2 and 8)
Blood is drawn from an experienced phlebotomy team using a butterfly needle and syringe and placed in microcontainers for transport to the laboratory for processing into serum and plasma and storage at −80°. In the case of laboratory studies to assess the safety of nicotinamide administration (performed on the first 100 children enrolled), a whole blood specimen is immediately tested in the clinical laboratory at Haydom Lutheran Hospital for complete blood count (CBC).

### Stool (months 6, 9, 12, 15 and 18)
Stool is collected in diapers that are provided just prior to the stool collection date. Mothers or caregivers are instructed to collect approximately 3–4 spoonfuls of stool and place it within 1 hour into a labelled stool container, seal in a plastic bag and place inside a transport box with cold packs. The stool collection will be timed prior to a planned study visit, such that the field-worker can collect the sample and the laboratory process the specimen shortly thereafter. Stool samples are then stored at −80°. A subset of specimens, prior to freezing will be cultured for *E. coli* and screened for antibiotic resistance (AMR). This is performed as surveillance for safety and will be performed on a randomly selected 25 participants per intervention allocation group at 12 and 15 months.

### Urine (months 6, 12 and 18)
Urine is collected by placing a bag over the child's genitalia at the beginning of a study visit. The child is given adequate hydration, via breastfeeding if possible. Collected urine is then transferred in a sterile container that is stored on ice prior to transport to the research site and then stored at −80°.

### Saliva (maternal and infant at month 12)
Maternal and child saliva is collected into a phial containing storage reagent and stored at −80° for genomic assessment.

### Breath (subset of participants at months 6, 9 and 12)
Glucose hydrogen breath testing is performed in a subset of children (the last 400 children enrolled) as an assessment of small intestine bacterial overgrowth (SIBO),[32]

which has been associated with intestinal inflammation and linear growth shortfall.[33] Children are fasted for 2 hours prior to the initiation of the breath test. They are then given 1 g/kg actual body weight of a glucose dissolved in a 1 g/5 mL solution. Following ingestion of the standardised glucose solution, participants have breath collected via the Quintron paediatric breath collection bag system every 20 min for 2 hours. This non-invasive procedure involves the child breathing into an anaesthesia mask connected to a one-way flutter valve and collection bag. Breath samples are extracted from the collection bag using a syringe and stopcock and placed into specialised vacutainer glass vials. These vials are then transported to our research facility at Haydom Lutheran Hospital for assessment.

## Laboratory procedures and storage
On receipt of the samples listed above, the laboratory team at the research site prepares aliquots of samples for storage at −80° for local testing or shipping to additional testing facilities as shown below.

## Breast milk
### Rationale
During the first 6 months of participation, the only means of the child receiving intervention is through maternal delivery of nicotinamide in breast milk. However, nicotinamide in vivo is both derived from tryptophan and converted to multiple other vitamers.[34] As such, we will assess content of nicotinamide, vitamers and tryptophan in breast milk at two time points during the intervention.

Maternal breast milk samples collected at months 1 and 5 will be tested at the United States Department of Agriculture laboratory at the University of California–Davis. Planned LC-MS analyses includes nicotinamide, thiamin, riboflavin, flavin adenine dinucleotide, pyridoxal, pyridoxine, biotin and pantothenic acid concentrations and on a random subset (n=600 samples) niacin vitamer and free and total tryptophan concentrations.

## Blood
### Rationale
Blood testing will include assessment of multiple measures both to verify safety of the interventions and interrogate the effects of the intervention on physiological processes related to linear growth. Safety assessment includes testing of general tests of health with basic metabolic panel (BMP), liver function and enzyme testing and CBC. This will assess for differences between intervention groups in measures of kidney, liver and haematopoetic health. Additional tests to interrogate the physiological effects of the intervention include assessment of metabolic effects (using metabolomics) and assessment of levels of factor related to growth (insulin-like growth factor-1 (IGF-1) and collagen X).

### Complete blood count (CBC)
To assess general safety of the nicotinamide dosing in the first 100 infants at months 2 and 8. CBC is performed

at Haydom Lutheran Hospital clinical on a Swelab Alfa haematology analyzer (Boule Diagnositics, Stagna, Sweden; this machine has quality control assessed daily).

### Basic metabolic panel (BMP), aminotransferases and bilirubin

To assess general safety of the nicotinamide dosing in the first 100 infants at months 2 and 8, additional laboratory measures will be performed on serum samples at the Kilimanjaro Christian Medical Center. BMP, bilirubin, alanine aminotransferase and aspartate aminotransferase will be assessed using a Cobas Integra 400 Plus biochemistry analyzer (Roche Diagnostics, Germany), according to the manufacturer's instructions.[35]

### Insulin-like growth factor-1 (IGF-1) and high-sensitivity C reactive protein (hsCRP)

To interrogate these interventions for changes in growth factor and low-grade systemic inflammation, respectively, IGF-1 and hsCRP will be tested at 12 and 18 months at the UVa Center for Research in Reproduction Ligand Core Laboratory using an Immulite 2000.

### Collagen X

We will assess for intervention-associated changes in collagen X (a biomarker of rapid skeletal growth) using an ELISA developed at Shriners Hospital for Children in Portland, OR and Oregon Health and Science University.[36]

### Metabolomics

The tryptophan-kynurenine-niacin pathway will be assessed at 12 and 18 months via metabolomics analyses of serum and plasma using $^1$H nuclear magnetic resonance (NMR) spectroscopy and ultra-performance liquid chromatography-mass spectrometry (UPLC-MS) at Imperial College, London. Assessments include the following: (1) metabolites of the tryptophan-kynurenine-niacin pathway to assess for (among other items) changes in the tryptophan:kynurenine ratio in the setting of nicotinamide treatment[24]; (2) plasma bile salts, which have implications for risk of infection[37]; (3) untargeted metabolomics analysis, providing information on the global metabolic status of the individual including pathways related to the metabolic activity of the gut microbiota and their biochemical interactions with the host[38] and untargeted UPLC-MS based lipidomics to assess for differences in lipid metabolism by nutritional status.

### Stool

Stool testing will include assessment of the following:

### Microbiome

Stool bacterial content will be assessed at in a subset of 200 participants at the University of Maryland. Total stool DNA will be isolated by bead beating method. A library will be created for each specimen. The libraries will be multiplexed and sequenced by Illumina HiSeq. The resulting reads will be cleaned, binned by specimen ID, separated in operational taxonomic units (OTUs) by DNAclust[39]

and compared using metagenomeseq. Metagenomeseq normalises the data and introduces a correction factor to account for undersampling of OTUs. The normalised data will be compared using ORs and multiple t-tests, with p values adjusted for multiple testing.

### Taqman Array Cards (TAC)

Pathogen burden will be assessed using TAC cards (figure 2) as performed previously.[40–42] These tests will be performed in the laboratory at Haydom Global Health Research Center using. Briefly, stool samples undergo nucleic acid extraction via the QIAamp Fast Stool DNA Mini kit (Valencia, California, USA). Nucleic acids are mixed with AgPath-ID One-Step RT-PCR buffer (Thermo Fisher) and nuclease free water to a 100 ul final volume and run on a custom-designed TAC.[40]

### Antibiotic resistance

We will perform stool bacterial culture on a subset of participants to monitor for development of antimicrobial resistance, which is a valid concern of any mass antimicrobial administration. Briefly, stool will be cultured on MacConkey agar and five lactose fermenting *E. coli* colonies pooled and tested for AMR using disc diffusion to screen for ESBL, AmpC, carbapenemase and susceptibility to antibiotics commonly in use locally. Azithromycin susceptibility will be tested by E-Test.[43]

### Myeloperoxidase

As an assessment of stool inflammation, myeloperoxidase will be assessed at months 6, 12 and 18. Testing will be performed at the Global Health Research Center laboratory in Haydom using a commercially available assay (Alpco, Salem, New Hampshire, USA).

### Urine

Urine samples at 6, 12 and 18 months will assessed alongside serum and plasma samples at Imperial College, London for untargeted metabolomics analysis by NMR spectroscopy and UPLC-MS, as described above.[38]

### Saliva

For a more complete assessment of these interventions on intestinal inflammation in this setting, we will test mothers and children for secretor status of fucosyltransferase 2 (*FUT2*), a regulator of blood group antigens on the intestinal mucosa, as non-secretor status has been associated with infections[44] and other forms of colitis.[45] DNA for mothers and children will be extracted from salivary samples at UVa and *FUT2* genotype will be determined by dot blot assays in a hierarchical method based largely on Lewis Type a and b antigens. Lewis double negatives (5%–7%) are not able to be characterised as for their secretor status and so will be typed for A, B and H antigens to allow for phenotypic characterisation of all samples.[46]

### Breath

To interrogate these interventions for changes in SIBO, a pathophysiological process associated with poor linear

growth, we will assess hydrogen and methane quantity of breath samples in a subset of 200 participants. Breath samples collected in glass vials will be extracted and analysed using a Quintron BreathTracker SC breath chromatograph (Quintron, Milwaukee, Wisconsin, USA).[33] Data on both hydrogen and methane production will be collected.

### Data management and confidentiality

Data are collected onto paper forms, which are then quality checked before being double-entered by two different study team data clerks independently at the research site into a secure web-based system (Multi-Schema Information Capture, MuSIC) managed by UVa Department of Public Health Sciences. Any discrepancies in this double-data entry are reconciled by assessing the original paper form.

### Data analysis

#### Primary endpoint

The primary outcome for the overall study is LAZ at 18 months. For each of the intervention domains we will compare this outcome as a t-test between those randomised to receive the intervention versus those who did not. This will be performed using a modified intent-to-treat analysis, assessing participants who complete the study and have a measurement of LAZ at 18 months.

#### Secondary endpoints

Each of the intervention domains has its own set of secondary outcomes as well (see table 2), and these will be compared between those who receive the intervention and those who do not. This will initially be performed using the same modified intent-to-treat analysis assessing those with the individual outcome of interest. This will be performed as t-tests for continuous outcomes and $\chi^2$ tests for binary outcomes.

Secondary analysis for all endpoints (including LAZ at 18 months) will also be performed using per-protocol treatment, defined as participants meeting the following criteria:

► Breastfed through age 6 months.
► Received all doses of azithromycin.
► Received initial dose of nitazoxanide.
► Received at least 50% of nicotinamide doses as measured by pill and sachet counting.
► Have the outcome of interest measured.

### Blinding

The study is being performed in a double-blinded manner. Individual study medications were purchased from the same manufacturer as their corresponding placebo and placebos were designed to be similar in appearance and taste. Study personnel received the medications and placebos merely labelled as 'A' and 'B,' with the key to this allocation provided to one individual at

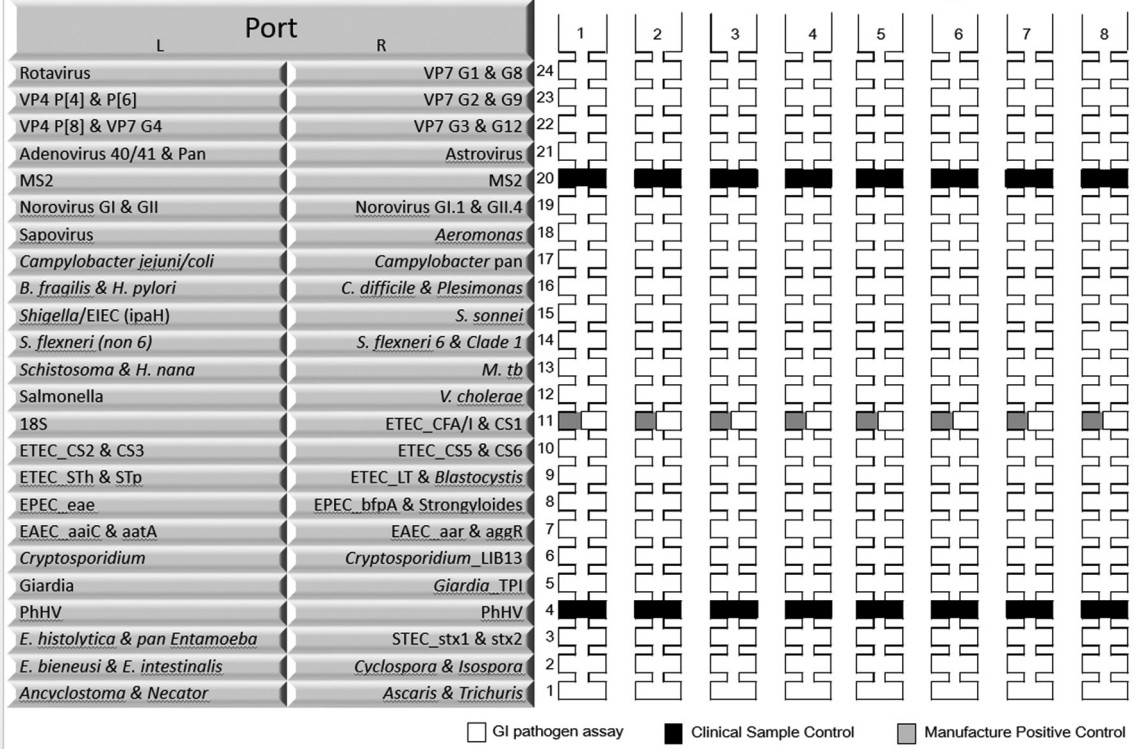

**Figure 2** TAC schematic. Burden of intestinal pathogens will be assessed in stool samples from participants at multiple time points using TAC. TAC, Taqman Array Card.

**Table 2**  Primary and secondary outcome measures by study intervention

| Outcome | Timing (age of child in months) | Antimicrobial intervention | Nicotinamide intervention |
|---|---|---|---|
| **Primary outcome** | | | |
| LAZ | 18 | X | X |
| **Secondary outcomes** | | | |
| Proportion of stunting (LAZ<–2) | 18 | X | X |
| WAZ and WLZ z-scores | 18 | X | X |
| Metabolomic assay of gut microbial metabolism and alterations to tryptophan-niacin and other pathways | 6, 12, 18 | X | X |
| Tryptophan:kynurenine ratio and other pathway metabolites in serum | 12, 18 | X | X |
| IGF-1 at 12 and 18 months | 12, 18 | X | X |
| High-sensitivity C reactive protein at 12 and 18 months | 12, 18 | X | X |
| Stool myeloperoxidase at months 6, 12 and 18 months | 6, 12, 18 | X | X |
| Prevalence of enteric pathogens | 6, 6.5, 12, 12.5, 18 | X | X |
| Change in microbiota (by traditional sequencing) (subset of participants) | 6, 6.5, 12, 18 | X | X |
| Change in small intestinal bacterial overgrowth as assessed by exhaled increased hydrogen following ingestion of sugar (subset of participants) | 6, 12, 18 | X | X |
| Difference in breast milk composition (nicotinamide, tryptophan, amino acids) | 1, 5 | | X |
| Symptomatic diarrhoea | | X | X |
| Hospitalisation and all-cause mortality | 18 | X | X |
| Cognitive outcomes MDAT assessment and the Bayley Scales of Infant and Toddler Development | 18 | X | X |

IGF-1, insulin-like growth factor-1; LAZ, length-for-age Z-score; MDAT, Malawi Development Assessment Tool; WAZ, weight-for-age; WLZ, weight-for-length.

UVa not otherwise associated with the study. Blinding will be maintained throughout the study unless adverse events occur in significantly disproportionate amounts by intervention allocation status. Allocation status will otherwise only be revealed following completion of collection of data and samples.

### Data and safety monitoring

A data and safety monitoring board has been formed to review data regarding safety of these interventions. This board contains four experts and physicians from Tanzania, the USA and the UK with specialties in gastroenterology, global health, paediatrics and statistics to assist with reviewing data related to adverse events, health outcomes and surveillance cultures evaluating for AMR. Adverse event data are reported monthly by treatment allocation group (in a blinded fashion). The DSMB makes recommendations regarding ongoing safety of the interventions and any need to be unblinded due to differences in adverse events between treatment allocation group and can recommend discontinuation of any of the intervention arms.

### Statistical power

The primary outcome is difference between intervention groups in LAZ at 18 months. The sample size was calculated to provide sufficient power for testing the main effects using separate t-tests for each of the two interventions in this 2×2 factorial design, using a SD of 1.03 HAZ at 18 months. If each main effect were tested at the 5% levels, as is customary in the analysis of factorial studies,[47 48] with 270 participants per group, there is 80% power for a difference in LAZ of 0.176. This is within the range of improvement of an increase of 0.16 seen in a prior study after 12 months of micronutrient supplementation,[49] though in Haydom the growth and deficiencies are more severe than most areas, thus the potential for observing a greater effect size. The interaction between main effects of these interventions will then be estimated and tested as a secondary analysis using a 2-way analysis of variance (ANOVA). If we were to take a more conservative approach and use a significance level of 2.5% for each of the main effects, with 270 subjects per group, we would still have 80% power to detect an average LAZ increase of 0.193. Finally, if we were to adjust the significance to include the test for the interaction, the F-test for the main effect in a 2-way ANOVA has 80% power with a two-sided significance level of 1.67%, when the main effect of the intervention is to increase the mean LAZ by 0.203. All of these are reasonable changes in LAZ to observe for in the Haydom population, given the considerable demonstrated mean deficit in growth. Adjusting for 10% dropout, the total

sample size required is 1188. Within our recruitment radius >2507 births occur yearly. We will recruit during a 12-month window. In our extensive research experience at the Haydom MAL-ED site, we observed 0% refusal to participate and a high 81% retention rate by 4 years of follow-up during MAL-ED (and, thus, a study of shorter duration may have improved follow-up). Based on this, we anticipate a high rate of recruitment (>90%) and 18-month retention (>90%). Therefore, accounting for dropout, we anticipate achieving 1188 mother/child dyads divided between intervention and control, leaving us adequate power to determine differences in LAZ.

## Statistics
In addition to use of ITT for the central analysis (t-tests) and the standard 2-way ANOVA to assess main effects and the interaction, we will perform several sensitivity analyses to assess for influences from children who dropped out or died during the study. These will include use of linear mixed-effects models based on repeated measures of anthropometric measures (adjusted for baseline LAZ), assessments of last-measurement carried forward and assumptions about those who dropped out being taller or shorter than the mean. Additional secondary analyses will include assessing HRs for events such as mortality and hospitalisation, including time to event analysis.

## Study timeline
Enrolment began 5 September 2017 and is planned to continue through 5 September 2018. Children will be followed until age 18 months, resulting in a planned completion in February 2020. Blinding will be maintained until all final study visits are completed. The primary analysis will be performed once data collection is complete for the variables involved. To reduce intra-assay variability, the majority of laboratory testing will be batched and performed once complete samples are present.

## Potential challenges and limitations
Potential challenges include adequate recruitment of target participant numbers within the 1-year study window. Should this be true, we will consider extending the recruitment window. Maintaining adherence of study medication, particularly the daily doses of nicotinamide to breast-feeding mothers during months 0–6, and infants during months 6–18, may be a challenge in a setting in which few individuals are accustomed to taking daily medication or nutritional supplements. To protect against adherence challenges, we performed a pilot study beforehand, identifying potential barriers to adherence and resulting in steps to remind mothers frequently during the first month of the study, with follow-up home visits by the field team and by local community healthcare workers.

## ETHICS AND DISSEMINATION
### Regulatory authorities
ELICIT has received approval from Tanzania's National Institute of Medical Research (NIMR, reference number HQ/R.8a/Vol.IX/2424, approved 3 March 2017), the University of Virginia Health Sciences Research Institutional Review Board (HSR-IRB, reference #19465, approved 19 January 2017) and the Tanzanian Food and Drug Administration (TFDA, reference number TFDA0017/CTR/0005/02, approved 5 July 2017). All research procedures are conducted in accordance with the Declaration of Helsinki and according to International Conference on Harmonisation Good Clinical Practice guidelines. All participant mothers provide informed consent. An external monitor (FHI360) evaluated the site prior to initiation of the study and performs yearly site visits and assessments thereafter.

### Patient and public involvement
Patients and the public were not directly involved in the planning of this study. Results will be communicated to participants via community meetings with local leaders.

### Dissemination
This trial has been registered at ClinicalTrials.gov (#NCT03268902). Scientific reports resulting from this study will be made publicly available. Following completion of data accrual, data will be made publicly available in an online database. Data will be presented at major conferences such as those of the American Society of Tropical Medicine and Hygiene, the Pediatric Academic Society, the Endocrine Society and the African Academy of Sciences. Scientific articles will be published with open-access availability at major scientific journals.

**Author affiliations**
[1]Department of Pediatrics, University of Virginia, Charlottesville, Virginia, USA
[2]Department of Medicine, University of Virginia, Charlottesville, Virginia, USA
[3]Department of Global Health and Primary Care, University of Bergen, Bergen, Norway
[4]Department of Surgery & Cancer, Imperial College of London, London, UK
[5]Division of Infectious Disease, Children's Hospital of Richmond at Virginia Commonwealth University, Richmond, Virginia, USA
[6]Haydom Global Health Research Centre, Haydom Lutheran Hospital, Haydom, Tanzania

**Acknowledgements** Romark donated the nitazoxanide.

**Contributors** MDD, JAP-M, RJS, JMM, JG, ES, ERH and EM conceived of this trial and developed this study protocol. EM is the Principal Investigator. MDD is the UVa Principal Investigator. SJ is the Clinical Research Coordinator. AWW is the UVa Project Manager. JAP-M, JG and EH developed procedures related to stool testing for pathogens and antibiotic resistance. JS developed procedures related to metabolomics testing. JRD developed procedures related to testing for small intestinal bacterial overgrowth. JMM developed procedures related to the breastfeeding support of mothers for community, field and nursing staff. RJS and ES developed procedures related to developmental testing. JG oversees all sample processing and laboratory procedures in Haydom. All authors contributed to the development of this manuscript and/or study procedures and to reading and approving the final version for publication.

**Funding** This work was supported by the Bill and Melinda Gates Foundation, OPP1141342.

**Competing interests** None declared.

**Patient consent** Not required.

**Ethics approval** National Institute for Medical Research, Tanzania; IRB, University of Virginia; Food and Drug Administration, Tanzania.

**Provenance and peer review** Not commissioned; externally peer reviewed.

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
