## [Reviewer comments · BMJ Open]

ARTICLE DETAILS

TITLE (PROVISIONAL)	Early Life Interventions for Childhood Growth and Development in Tanzania (ELICIT): A Protocol for a Randomized Factorial, Double-Blind, Placebo-Controlled Trial of Azithromycin, Nitazoxanide and Nicotinamide
AUTHORS	DeBoer, Mark; Platts-Mills, James; Scharf, Rebecca; McDermid, Joann; Wanjuhi, Anne; Gratz, Jean; Svensen, Erling; Swann, Jon; Donowitz, Jeffrey; Jatosh, Samwel; Houpt, Eric; Mduma, Estomih

VERSION 1 – REVIEW

REVIEWER	Jean-François Rossignol MD PhD Romark Laboratories, LC, Tampa Florida 33607 USA
REVIEW RETURNED	09-Feb-2018

GENERAL COMMENTS	This very interesting protocol is using a modified intent to treat (ITT) analysis of the primary endpoints. The protocol does not give enough details about what would be a traditional intend-to-treat analysis compared to the modified ITT. It illustrates a trend in clinical trials to use this type of patient population instead of the pure ITT group of patient.
---

REVIEWER	Patricia Pavlinac University of Washington, USA
REVIEW RETURNED	13-Feb-2018

GENERAL COMMENTS	This is an exciting (and ambitious) randomized controlled trial addressing an important outcome (linear growth) for which few interventions exist. The authors provide excellent rationale for the study and the interventions being tested. However there are a few analytic and methodological details that could be better detailed in the protocol: Pg 5 line 18/19: The statement “70% rate of stunting by 18 months of age” is unclear. Is this 70% of children were stunted at 18 months of age or that the incidence rate of stunting was 70/100-child years. If the former, I would suggest calling it a prevalence rather than a rate or using language that distinguishes the two. Although outside of the scope of this introduction, that prevalence is strikingly high. I’m curious how this population may be unique from other populations in East Africa (perhaps higher than average HIV status). That the nicotinamide is being administered to the mothers during the first 6 months of the infant’s life is an interesting concept. Why was this chosen instead of giving directly to the child? Given that the original MAL-ED data (MAL-ED PLOS MED 2017), reported 44 days
---

of exclusive breastfeeding of this site, are the authors concerned that children will not receive adequate amounts of this intervention? I would expect there being a difference in nicotamide transfer to the infant between infants being exclusively vs. non-exclusively breastfed. The secondary per-protocol analysis based on breastfeeding status may need to disaggregate children who are exclusively vs. non-exclusively vs. not breastfed at all. Relatedly, if a mother stops breastfeeding completely during the first 6 months of the infants' life, or dies, will nicotamide be administered through sachets?

That nicotamide is expected to be given daily to children from 6 months of age to 18 months of age seems extremely ambitious, given that even 14-day courses of interventions, such as zinc, are rarely adhered to. Although the authors raise this challenge in the limitations, including reminders and home visits, it would be helpful to discuss why such a long course is necessary in the introduction. Also in the per-protocol analysis, it says that you will examine among those who received >50% of pills, does this also include sachets?

Although I recognize the protocol has started so would be too late to change, I'm curious why the authors chose to wait to administer the azithromycin until 6 months of age. Is this due to safety concerns or based on biological plausibility of effect? In reviewing the LAZ growth curve from the Tanzania site of MAL-ED (MAL-ED PLOS MED 2017), it appears that children began their downward LAZ trajectory prior to 6 months of age which begs the question of whether those first 6 months of life would not in fact be an ideal time to treat (or prevent) enteric infections.

Due to the somewhat complicated factorial design and intervention time-points, I would suggest adding a figure that shows the various points in the study (from enrollment at day 14 through the last follow-up visit at 18 months, including intervention time points (and whether the intervention is administered to the mother or the child) and follow-up time points.

The protocol states that a subset of stool samples will be cultured for E.coli and antibiotic susceptibility testing performed however there is no specification of which timepoints this will be assessed (and this outcome is not listed in Table 1).

There are a number of samples being collected and assays being performed which will prove useful for understanding mechanism of effect. However there is no specification on how subsets will be chosen for these samples, with these be a random subset of children, children selected based on outcomes, or some other selection criteria?

HIV status of the child and mother are not mentioned as a potential covariates but may be important for interpreting variability in effects. Since blood is already being collected, the authors may want to consider adding this assay (with ethical approvals).

The statistical analysis section is nested in the study power section and should be removed as a separate section. It's also missing some key methodological details which should either be addressed in the protocol or made clear where a more detailed statistical analysis plan can be found. Such pre-specification is recommended

	in the Spirit guidelines and will protect the investigative team from critiques posed after the results are published. These include:  1) How will deaths be dealt with in the primary analysis? Given the incredibly high stunting prevalence in this population, it's likely that there will be a number of deaths and could lead to bias in the overall effects if risk of dying was influenced by the interventions (which may be particularly plausible for azithromycin given the Ethiopia MDA trial mentioned in the introduction). 2) Primary outcome is listed as LAZ at M18 (pg 19 line 3) and difference in LAZ at 18 months (pg 20, line 14). These are slightly different outcomes, with differing power, and so important to pre-specify which is the primary outcome to be used (comparing means of change or means of LAZ). There is no clear statement about how LAZ (mean change or mean in absolute LAZ) will be analyzed, although perhaps it's implied in the statement on "2-way ANOVA to assess main effects." If this is the primary analysis, I would making it more clear. 3) If missing the M18 FU, will the child be excluded from the analysis even if he/she had a length measurement from an earlier visit? 4) Pg 21, line 9-13 lists a number of covariates. For the primary analysis, I would presume these are not going to be included, but would be included in analyses secondary to the primary analysis. Should make this distinction clear if that is the case. 5) The authors state the analysis will be a modified intention-to-treat but the details on what is modified is not clear. Is it modified in the sense that only those with a M18 visit are included or are there other post-randomization exclusions that are occurring for the primary analysis? 6) I would suggest moving the statement about all endpoints being analyzed using a per protocol analysis to a separate paragraph (pg 19line 24) so as to separate the types of analyses being performed from the primary and secondary outcomes. 7) There are a number of secondary endpoints, most with measures at multiple time-points. There is no mention of how the multiple timepoints will be handled in the analysis nor whether the authors will account for the high likelihood of erroneously statistically significant effects given the number of hypothesis tests being conducted (ie will there be adjustment for multiple comparisons?). 8) Some of the secondary outcomes would be best analyzed using a time-to-event analysis (rather than logistic regression) such as death or hospitalization given the long follow-up period if the authors are not confident in the high retention rate. 9) In the statistical power section, page 20, line 20-22, please list the assumed standard deviation for the observed difference in LAZ of 0.176. This would be useful to compare actual SD when results are published. 10) In the Data Safety section, there is no discussion of the plan for interim analyses and stopping guidelines. Given this is outlined in the Spirit guidelines as a requirement, should either include those specifics in the protocol or reference the DSMC charter where this information could be found.
--	--

VERSION 1 – AUTHOR RESPONSE

Reviewers' Comments to Author:

Reviewer: 1

Reviewer Name: Jean-François Rossignol MD PhD Institution and Country: Romark Laboratories, LC, Tampa Florida 33607, USA Competing Interests: None declared

This very interesting protocol is using a modified intent to treat (ITT) analysis of the primary endpoints. The protocol does not give enough details about what would be a traditional intend-to-treat analysis compared to the modified ITT. It illustrates a trend in clinical trials to use this type of patient population instead of the pure ITT group of patient.

RESPONSE: Thank you. The primary intent-to-treat analysis is modified in the sense that we are only assessing those individuals who have a measured value for the primary outcome (Primary endpoint, page 19, paragraph 4, line 3). Based on Reviewer 2's recommendation, we now include a "Statistics" section that discusses that we will also perform sensitivity analyses that may be considered more traditional ITT (Statistics, page 23, paragraph 2, line 1).

Reviewer: 2

Reviewer Name: Patricia Pavlinac

Institution and Country: University of Washington, USA Competing Interests: None declared

This is an exciting (and ambitious) randomized controlled trial addressing an important outcome (linear growth) for which few interventions exist. The authors provide excellent rationale for the study and the interventions being tested. However there are a few analytic and methodological details that could be better detailed in the protocol:

1. Pg 5 line 18/19: The statement "70% rate of stunting by 18 months of age" is unclear. Is this 70% of children were stunted at 18 months of age or that the incidence rate of stunting was 70/100-child years. If the former, I would suggest calling it a prevalence rather than a rate or using language that distinguishes the two. Although outside of the scope of this introduction, that prevalence is strikingly high. I'm curious how this population may be unique from other populations in East Africa (perhaps higher than average HIV status).

RESPONSE: We have revised this sentence to read, "Children in Haydom followed in MAL-ED had a prevalence of stunting of 70% at 18 months of age" (Introduction, page 5, paragraph 1, line 7). Regarding the reason for the high prevalence of stunting, Haydom is a very rural area with a relatively short growing season and a low degree of development. Fortunately, it has a very low prevalence of HIV infections (<2%) and malaria (these data and more about Haydom are now provided in a supplemental section on "Setting, page 8, paragraph 3"). MAL-ED identified multiple pathogens that contribute to the poor growth. While not specifically identified in MAL-ED, it is difficult to imagine that food scarcity does not play a role in increasing risk of stunting.

2. That the nicotinamide is being administered to the mothers during the first 6 months of the infant's life is an interesting concept. Why was this chosen instead of giving directly to the child? Given that the original MAL-ED data (MAL-ED PLOS MED 2017), reported 44 days of exclusive breastfeeding of this site, are the authors concerned that children will not receive adequate amounts of this intervention? I would expect there being a difference in nicotinamide transfer to the infant between infants being exclusively vs. non-exclusively breastfed. The secondary per-protocol analysis based on breastfeeding status may need to disaggregate children who are exclusively vs. non-exclusively vs. not breastfed at all. Relatedly, if a mother stops breastfeeding completely during the first 6 months of the infants' life, or dies, will nicotinamide be administered through sachets?

RESPONSE: We have addressed all of these important issues by adding to the text as follows: "During the first 6 months of life, this is delivered to the child via breastmilk. This is performed because of the lack of availability of a liquid preparation of nicotinamide and to emphasize the importance of exclusive breast feeding (i.e., that intake beyond breast milk is not needed during the first 6 months of life)... Because the prevalence of exclusive breastfeeding was low in the area around Haydom during MAL-ED, the current study includes a breast-feeding encouragement and support initiative, utilizing nurses who have received specialized training in breast feeding; these nurse work with the local community healthcare workers in home visits to participating families in the first 6

months of the study. Their message is that exclusive breast feeding is better for mother and baby (highlighting a variety of reasons) and that if the family chooses not to breast feed, that providing breast milk before any complimentary foods will allow the child to receive a greater amount of the needed nutrition—including a greater amount of the nicotinamide ingested by the mother. The lack of intent to breast feed is an exclusion criterion at the time of enrollment; however, if the mother dies or discontinues breast feeding, the child subsequently would not receive nicotinamide/placebo until age 6 months.” (Intervention, page 11, paragraph 2, line 2.)

3. That nicotinamide is expected to be given daily to children from 6 months of age to 18 months of age seems extremely ambitious, given that even 14-day courses of interventions, such as zinc, are rarely adhered to. Although the authors raise this challenge in the limitations, including reminders and home visits, it would be helpful to discuss why such a long course is necessary in the introduction. Also in the per-protocol analysis, it says that you will examine among those who received >50% of pills, does this also include sachets?

RESPONSE: We have added the following to the Introduction: “However, because growth is a long-term process and nicotinamide is a water-soluble vitamin without a mechanism for on-going storage, we reasoned that an intervention with nicotinamide would need to be given over a long period of time.” (Introduction, page 7, paragraph 1, line 11). We agree that it is critical for the field workers and community health workers to continue to encourage adherence. We have added that the per-protocol analysis also includes counting of un-used sachets.

4. Although I recognize the protocol has started so would be too late to change, I’m curious why the authors chose to wait to administer the azithromycin until 6 months of age. Is this due to safety concerns or based on biological plausibility of effect? In reviewing the LAZ growth curve from the Tanzania site of MAL-ED (MAL-ED PLOS MED 2017), it appears that children began their downward LAZ trajectory prior to 6 months of age which begs the question of whether those first 6 months of life would not in fact be an ideal time to treat (or prevent) enteric infections.

RESPONSE: We agree that initiation at 3 months would have been reasonable. We have now included the following justification for initiation at 6 months: “This timing was selected (versus earlier initiation) because while children have enteropathogen carriage before 6 month in this area, carriage of azithromycin-susceptible pathogens such as Campylobacter are not highly prevalent, and the decline in LAZ is steeper in the 6-18-month window compared to the 0-6 month window” (Intervention, page 10, paragraph 3, line 6).

5. Due to the somewhat complicated factorial design and intervention time-points, I would suggest adding a figure that shows the various points in the study (from enrollment at day 14 through the last follow-up visit at 18 months, including intervention time points (and whether the intervention is administered to the mother or the child) and follow-up time points.

RESPONSE: Thank you—we have added a chart, as also suggested by the editor (Table 1).

6. The protocol states that a subset of stool samples will be cultured for E.coli and antibiotic susceptibility testing performed however there is no specification of which timepoints this will be assessed (and this outcome is not listed in Table 1).

RESPONSE: The antibiotic resistance testing is performed not as a secondary endpoint but as a safety assessment. This point and the timing of testing has not been included in the text: “This is performed as surveillance for safety and will be performed on a randomly-selected 25 participants per intervention allocation group at 12 and 15 months” (Specimen collection, page 15, paragraph 1, line 1).

7. There are a number of samples being collected and assays being performed which will prove useful for understanding mechanism of effect. However there is no specification on how subsets will

be chosen for these samples, with these be a random subset of children, children selected based on outcomes, or some other selection criteria?

RESPONSE: We have clarified in the text that the safety laboratory samples are performed on the first 100 children enrolled in the study (Specimen collection, page 14, paragraph 4, line 4), the small-bacterial overgrowth samples are performed on the last 400 children in the study (Specimen collection, page 15, paragraph 4, line 2), and the other samples are performed on a random sample without consideration to outcome.

8. HIV status of the child and mother are not mentioned as a potential covariates but may be important for interpreting variability in effects. Since blood is already being collected, the authors may want to consider adding this assay (with ethical approvals).

RESPONSE: While HIV infections are of clear importance in the health of individuals in sub-Saharan Africa, the area around Haydom has a very low prevalence of HIV infection in women of childbearing age (now stated in the text: Setting, page 8, paragraph 1, line 1). For the sake of privacy (and to not discourage participation), testing of HIV is not in our consent form, and we are thus not approved for its testing.

9. The statistical analysis section is nested in the study power section and should be removed as a separate section. It's also missing some key methodological details which should either be addressed in the protocol or made clear where a more detailed statistical analysis plan can be found. Such pre-specification is recommended in the Spirit guidelines and will protect the investigative team from critiques posed after the results are published. These include:

9.1 How will deaths be dealt with in the primary analysis? Given the incredibly high stunting prevalence in this population, it's likely that there will be a number of deaths and could lead to bias in the overall effects if risk of dying was influenced by the interventions (which may be particularly plausible for azithromycin given the Ethiopia MDA trial mentioned in the introduction).

RESPONSE: As described for Reviewer #1, the modified intent-to-treat analysis of the primary endpoint will not include those who do not have the measurement of LAZ at M18. However, we now clarify that we will perform several sensitivity analyses to further assess for the relationship between the interventions and growth (Statistics, page 23, paragraph 2, line 1). In the case of antimicrobial treatment, omission of participants who died during the study may more likely bias us against toward finding an effect of the intervention, if placebo-treated participants did not survive but did not grow as well during that time. Nevertheless, we will do these sensitivity analyses both examining effects if participants who died were shorter or taller than surviving participants.

9.2 Primary outcome is listed as LAZ at M18 (pg 19 line 3) and difference in LAZ at 18 months (pg 20, line 14). These are slightly different outcomes, with differing power, and so important to pre-specify which is the primary outcome to be used (comparing means of change or means of LAZ). There is no clear statement about how LAZ (mean change or mean in absolute LAZ) will be analyzed, although perhaps it's implied in the statement on "2-way ANOVA to assess main effects." If this is the primary analysis, I would making it more clear.

RESPONSE: We have attempted to clarify this by describing this is "difference between intervention groups in LAZ at 18 months." We have also clarified that this primary analysis is performed as t-tests for the antimicrobial and nicotinamide interventions separately. We have clarified use of the 2-way ANOVA as follows: "The interaction between main effects of these interventions will then be estimated and tested as a secondary analysis using a 2-way ANOVA." (Statistical power, page 22, paragraph 2, line 1.)

9.3 If missing the M18 FU, will the child be excluded from the analysis even if he/she had a length measurement from an earlier visit?

RESPONSE: Children missing M18 FU will not be included in the primary ITT but will be assessed in sensitivity analyses, including last-measurement carried forward. This is now mentioned in the Statistics section (Statistics, page 23, paragraph 2, line 1).

9.4 Pg 21, line 9-13 lists a number of covariates. For the primary analysis, I would presume these are not going to be included, but would be included in analyses secondary to the primary analysis. Should make this distinction clear if that is the case.

RESPONSE: These are not going to be listed in the primary analysis, which has been clarified to say: "For each of the intervention domains we will compare this outcome as a t-test between those randomized to receive the intervention vs. those who did not" (Primary endpoint, page 20, paragraph 4, line 1).

9.5 The authors state the analysis will be a modified intention-to-treat but the details on what is modified is not clear. Is it modified in the sense that only those with a M18 visit are included or are there other post-randomization exclusions that are occurring for the primary analysis?

RESPONSE: The intent-to-treat analysis is modified in the sense that we are only assessing those individuals who have a measured value for the primary outcome. There are no other exclusions from this analysis (Primary endpoint, page 20, paragraph 4, line 1).

9.6 I would suggest moving the statement about all endpoints being analyzed using a per protocol analysis to a separate paragraph (pg 19line 24) so as to separate the types of analyses being performed from the primary and secondary outcomes.

RESPONSE: Thank you—this has been done (Primary endpoint, page 21, paragraph 2, line 1).

9.7 There are a number of secondary endpoints, most with measures at multiple time-points. There is no mention of how the multiple timepoints will be handled in the analysis nor whether the authors will account for the high likelihood of erroneously statistically significant effects given the number of hypothesis tests being conducted (ie will there be adjustment for multiple comparisons?).

RESPONSE: We acknowledge that secondary endpoints have a more hypothesis-generating nature and are seen as such in the literature. While we are not adjusting for multiple comparison, we will acknowledge upon publication that these findings are secondary and thus could have occurred by chance alone.

9.8 Some of the secondary outcomes would be best analyzed using a time-to-event analysis (rather than logistic regression) such as death or hospitalization given the long follow-up period if the authors are not confident in the high retention rate.

RESPONSE: For the initial analysis of our secondary outcomes will be comparison of proportions using a chi-square test (now stated under "Secondary endpoints," page 21, paragraph 1, line 4). However, we agree that there is a place for hazard ratios of time-to-event for these outcomes and now state, "Additional secondary analyses will include assessing hazard ratios for events such as mortality and hospitalization, including time to event analysis" (Statistics, page 23, paragraph 2, line 5).

9.9 In the statistical power section, page 20, line 20-22, please list the assumed standard deviation for the observed difference in LAZ of 0.176. This would be useful to compare actual SD when results are published.

RESPONSE: This has been added: "The sample size was calculated to provide sufficient power for testing the main effects using separate t-tests for each of the two interventions in this 2 x 2 factorial

design, using a standard deviation of 1.03 HAZ at 18 months” (Statistical power, page 22, paragraph 2, line 3).

9.10 In the Data Safety section, there is no discussion of the plan for interim analyses and stopping guidelines. Given this is outlined in the Spirit guidelines as a requirement, should either include those specifics in the protocol or reference the DSMC charter where this information could be found.

RESPONSE: We have elected against interim analyses of our primary outcome. Because linear growth is harder to link to harm, we lacked appropriate criteria for stopping for benefit linked to the intervention. In addition, we lacked futility as a reason for stopping, since the multiple secondary outcomes allow us to assess for potential mechanisms that might explain underlying action (or lack of action) of the interventions. We are, however, following SAE and mortality by intervention group on an ongoing basis and if a difference in potential harm is noted in these outcomes, the DSMB can request unblinding for the committee and potentially stop the study. This is now stated as follows: “Adverse event data are reported monthly by treatment allocation group (in a blinded fashion). The DSMB makes recommendations regarding ongoing safety of the interventions and any need to be unblinded due to differences in adverse events between treatment allocation group and can recommend discontinuation of any of the intervention arms” (Data and safety monitoring, page 22, paragraph 1, line 5).

FORMATTING AMENDMENTS (if any)

Required amendments will be listed here; please include these changes in your revised version:

- Kindly re-upload FIGURE 2 with at least 300 dpi resolution.

RESPONSE: This has been done.

VERSION 2 – REVIEW

REVIEWER	Patricia Pavlinac PhD MS Assistant Professor Department of Global Health University of Washington Seattle WA USA
REVIEW RETURNED	23-Mar-2018

GENERAL COMMENTS	My comments have been sufficiently addressed. However I have a few follow-up points for the authors consideration. Response to 9.2 (re: LAZ at 18 months vs. change in LAZ between baseline & 18 months) If I’m understanding the responses correctly, the authors are pre-specifying that the primary outcome is LAZ at 18 months and mean 18 month LAZ will be compared across intervention arms. The authors may want to consider instead, change in LAZ (between baseline and 18 months) or ANCOVA (adjusting for baseline) rather than 18 month LAZ as accounting for baseline LAZ (either by considering change or by adjusting for baseline in the model) will increase precision (and thereby power) even is there is assumed balance in LAZ between randomization arms at baseline. Response to 9.3 (re: missing M18 LAZ) For sensitivity analyses, the carry-forward approach described for
--

	handling missing M18 LAZ may lead to significant bias if loss to follow-up occurred early in the follow-up period and if that LAZ (perhaps death) was associated with the intervention arm. Because the authors are collecting LAZ every 3 months, it seems counter-intuitive to not use all the LAZ data that is being collected using a modeling approach that accounts for repeated outcome measures (such as linear mixed effects models). Such an approach would enable censoring and could help identify critical time points at which improvements in LAZ (if any) seem to occur. M18 LAZ: There will inevitably be variability when children are seen for their M18 visit. Have the authors set pre-specified windows outside of which the measure will no longer be considered “18 months”? For example if a child is unavailable for the M18 visit but becomes available at M19, will the length measured 30-days after month 18 be used? The authors may consider including child anthropometric measures in Table 1.
--	---

REVIEWER	Jean-François Rossignol MD PhD Romark Laboratories LC USA
REVIEW RETURNED	29-Mar-2018

GENERAL COMMENTS	None but a little precision in the protocol: the unit used for weight/height are in kilogram, cm (metric system) and not the British Imperial system used in the UK and in the US.
--

VERSION 2 – AUTHOR RESPONSE

Reviewers' Comments to Author:

Reviewer: 1

Reviewer Name: Jean-François Rossignol MD PhD Institution and Country: Romark Laboratories LC, USA Competing Interests: None

None but a little precision in the protocol: the unit used for weight/height are in kilogram, cm (metric system) and not the British Imperial system used in the UK and in the US.

RESPONSE: Thank you—we have elected to continue with the metric units, as these are standard in Tanzania, where all of the measures are being performed. This improves interpretability for the participants and field team members.

Reviewer: 2

Reviewer Name: Patricia Pavlinac PhD MS
Institution and Country: Assistant Professor, Department of Global Health, University of Washington, Seattle WA, USA Competing Interests: None declared

My comments have been sufficiently addressed. However I have a few follow-up points for the authors' consideration.

1. Response to 9.2 (re: LAZ at 18 months vs. change in LAZ between baseline & 18 months) If I'm understanding the responses correctly, the authors are pre-specifying that the primary outcome is LAZ at 18 months and mean 18 month LAZ will be compared across intervention arms. The authors may want to consider instead, change in LAZ (between baseline and 18 months) or ANCOVA (adjusting for baseline) rather than 18 month LAZ as accounting for baseline LAZ (either by considering change or by adjusting for baseline in the model) will increase precision (and thereby power) even if there is assumed balance in LAZ between randomization arms at baseline.

RESPONSE: Thank you—we have chosen LAZ at 18 months as the primary outcome in part for comparison across other studies that may have different enrollment timing from ours (which is in the first 2 weeks of life). For example, WASH Benefits used LAZ at 24 months of age for a similar purpose. As you mention, the randomization and large n should balance enrollment measures between groups. Thus far, enrollment length has been well matched between intervention allocation groups.

2. Response to 9.3 (re: missing M18 LAZ)

For sensitivity analyses, the carry-forward approach described for handling missing M18 LAZ may lead to significant bias if loss to follow-up occurred early in the follow-up period and if that LAZ (perhaps death) was associated with the intervention arm. Because the authors are collecting LAZ every 3 months, it seems counter-intuitive to not use all the LAZ data that is being collected using a modeling approach that accounts for repeated outcome measures (such as linear mixed effects models). Such an approach would enable censoring and could help identify critical time points at which improvements in LAZ (if any) seem to occur.

RESPONSE: We agree with this and have revised our description of the secondary analysis to say, “We will perform several sensitivity analyses to assess for influences from children who dropped out or died during the study. These will include use of linear mixed-effects models based on repeated measures of anthropometric measures (adjusted for baseline LAZ), assessments of last-measurement carried forward and assumptions about those who dropped out being taller or shorter than the mean.” (Statistics, page 23, paragraph 2, line 3.)

3. M18 LAZ: There will inevitably be variability when children are seen for their M18 visit. Have the authors set pre-specified windows outside of which the measure will no longer be considered “18 months”? For example if a child is unavailable for the M18 visit but becomes available at M19, will the length measured 30-days after month 18 be used?

RESPONSE: Yes, we have pre-specified windows to be within 2 weeks of 18 months. After this point, final measurements will not be accepted. This is now stated: “All anthropometry measures for a given month (including the final visit) are able to be performed up to 14 days before or after the target date (which is based on the number of months after the child’s birthday)” (Anthropometry measures, page 12, paragraph 3, line 3).

4. The authors may consider including child anthropometric measures in Table 1.

RESPONSE: Thank you for noticing that omission—this has been added.